# Studies on the Inhibition Mechanism of Linalyl Alcohol against the Spoilage Microorganism *Brochothrix thermosphacta*

**DOI:** 10.3390/foods13020244

**Published:** 2024-01-12

**Authors:** Longteng Wang, Xing Liu, Wenxue Chen, Zhichang Sun

**Affiliations:** College of Food Sciences & Engineering, Hainan University, 58 People Road, Haikou 570228, China; wanglongteng@hainanu.edu.cn (L.W.); xliu@hainanu.edu.cn (X.L.); hnchwx@163.com (W.C.)

**Keywords:** linalyl alcohol, *Brochothrix thermosphacta*, membrane damage, DNA

## Abstract

The aim of this study was to investigate the bacterial inhibitory ability and mechanism of action of linalyl alcohol against *B. thermosphacta*. Linalyl alcohol causes the leakage of intracellular material by disrupting the cell wall and exposing the hydrophobic phospholipid bilayer, which binds to bacterial membrane proteins and alters their structure. In addition, linalyl alcohol causes cell membrane damage by affecting fatty acids and proteins in the cell membrane. By inhibiting the synthesis of macromolecular proteins, the normal physiological functions of the bacteria are altered. Linalyl alcohol binds to DNA in both grooved and embedded modes, affecting the normal functioning of *B. thermosphacta*, as demonstrated through a DNA interaction analysis.

## 1. Introduction

*Brochothrix thermosphacta* (*B. thermosphacta*) is a Gram-positive, rod-shaped, non-pathogenic, psychrotrophic bacterium considered to be one of the major spoilers of seafood and meat products; it is a microorganism that discolors meat and is often accompanied by the production of foul-smelling gases, leading to food waste and affecting the global economy and ecology [1,2,3,4]. In cold-smoked salmon, *B. thermosphacta* produces two compounds, namely 2-hexanone and 2-heptanone. These compounds are accountable for the development of an off-putting aroma similar to that of blue cheese [5]. The presence of *B. thermosphacta* in cooked and peeled shrimp can result in the development of undesirable odors such as strong butter, and buttermilk-like, sour, and nauseous off-odors. These odors are caused by the production of compounds such as 2,3-butanedione (diacetyl), 3-methyl-1-butanal, and 3-methyl-1-butanol by the bacteria [6].

Several interdiction approaches, including chemical preservatives, have been developed to avert the rise and spread of food spoilage microbes. However, the prolonged abuse of such chemical preservatives presents an inevitable risk to public health [7]. When dealt with inadequately, chemical preservatives can potentially be hazardous carcinogens or teratogens. Natural antibacterial agents can be used as food preservatives instead of the widely employed chemical preservatives [8,9]. The search for safe, natural antibacterial agents, such as plant extracts and essential oils (EOs), will thus be a primary priority in the future. Essential oils, odoriferous volatile byproducts of secondary plant metabolism, can be found in plentiful leaves and stems [10]. According to research, essential oils have qualities that are antibacterial, antifungal, anti-Schistosoma mansioni, and antiviral and that promote wound healing and wound contraction [11,12,13,14]. Linalyl alcohol (2,6-dimethyl-2,7-octadien-6-ol) is an aromatic monoterpene alcohol found in essential oils and is widely used in perfumes, cosmetics, household cleaners, and food additives [15]. The Food and Drug Administration (FDA) has approved it, and it is considered generally safe (GRAS) [16]. Recent studies have shown that linalyl alcohol has biological effects, including analgesic and anti-inflammatory characteristics [17,18], tumor chemotherapy-sensitizing effects [19], sedative and anxiolytic effects [20], insecticidal effects [21], and significant research and application value. 

In past studies, it has also been demonstrated that linalyl alcohol can exhibit excellent antimicrobial activity against a wide range of food-contaminating bacteria such as Shewanella putrefaciens, Pseudomonas aeruginosa, and Shigella [22,23,24]. But fewer studies have been conducted on how linalyl alcohol prevents *B. thermosphacta.* However, earlier studies have suggested that it can harm cell membrane integrity, produce macromolecules and intracellular enzyme efflux, avert protein synthesis, and impede cellular metabolism [22,23]. Additionally, linalyl alcohol can increase the membrane lipid oxidation-induced hyperpolarized cell membrane potential, decrease intracellular adenosine triphosphate (ATP) levels, and increase intracellular reactive oxygen species (ROS) concentrations [23,24,25,26]. As a result, the current work attempts to improve the antibacterial mechanism of linalyl alcohol by testing its disruptive and inhibitory effects on DNA, intracellular chemicals, and *Brochothrix thermosphacta* as a research topic.

## 2. Materials and Methods

### 2.1. Reagents and Test Bacteria 

#### 2.1.1. Reagents

Linalyl alcohol (≥99%), fluorescein diacetate, and 3,3-diethoxycarbocyanine iodine were purchased from Bohr Reagent Co., Ltd. (Shanghai, China). 2′,7′-Dichlorofluorescein diacetate was purchased from Yuanfan Biotechnology Ltd. (Shanghai, China). A bacterial genomic DNA kit and a bacterial total protein extraction kit were purchased from Sangon Biotech (Shanghai, China). 

#### 2.1.2. Bacterial Strains and Culture Conditions

Brochothrix thermosphacta strain BMZ066754 was provided by the China Agricultural Strain Preservation Center (Beijing, China). The strain was amplified via incubation at 26 °C for 2–3 d in a Nutrient Agar (NA) medium.

### 2.2. Research on Bacteriostatic Activity

#### 2.2.1. Determination of Minimum Inhibitory Concentration (MIC) of Linalyl Alcohol

The minimum inhibitory concentration (MIC) of linalyl alcohol was determined using the two-fold dilution method [27]. *Brochothrix thermosphacta* was incubated under linalyl alcohol stress at different concentrations (0.375, 0.75, 1.5, 3, 6, and 12 mL/L) for 24 h at 26 °C, and the lowest concentration at which there was significant growth inhibition after incubation was determined to be the MIC.

#### 2.2.2. Growth Curve

UV spectrophotometry was used to determine *Brochothrix thermosphacta*’s antibacterial kinetic assay [28]. Here, 2% (*v*/*v*) suspensions of the logarithmic growth phase were aspirated and injected into a lysogeny broth (LB) medium. Linalyl alcohol was added to the LB liquid medium to reach final concentrations of 1 MIC and 2 MIC. An equal amount of sterile water was used as a blank control. 

### 2.3. Mechanism of Action of Linalyl Alcohol on B. thermosphacta Cell Membranes

#### 2.3.1. Surface Hydrophobicity

Surface hydrophobicity was analyzed using the ANS fluorescent probe method to analyze the cell wall integrity of *B. thermosphacta*. Referring to the technique of Thennarasu et al. [29], the bacterial suspension (2.5 mL) and ANS solution (0.5 mL) were combined in a reaction tube, resulting in a final concentration of 6.67 mmol/L, and mixed using oscillations. Then, fluorescence spectra were measured using an F-4500 Fluorescence Spectrometer (Hitachi Corporation, Tokyo, Japan) with emission wavelengths ranging from 400 to 600 nm, an excitation wavelength of 380 nm, and excitation and emission slit widths of 5 nm.

#### 2.3.2. Crystal Violet and Fluorescein Diacetate Staining

The investigation of the mechanism of action of linalyl alcohol on the cell membrane was carried out via crystal violet staining [30]. The logarithmic-phase bacterial solution was taken, and different concentrations of the linalyl alcohol solution were added. After 6 h of incubation, the bacterial solution was taken for centrifugation and washed with sterile PBS, and the obtained bacterial body was mixed with a 0.1% crystal violet solution and incubated away from light. The supernatant was centrifuged, and its OD_590nm_ value was measured.

Bacterial membrane damage was further verified using the FDA staining assay. The fluorescence intensity of FDA can reflect the integrity and permeability of the bacterial cell membrane. Based on the methodology proposed by Gu with appropriate adjustments [31], different concentrations of the linalyl alcohol solution were added for incubation, the bacteria were collected and washed twice with PBS, and the supernatant was discarded and incubated with 250 μL of an FDA-acetone solution (2 mg/mL) for 20 min at room temperature. The average fluorescence intensity was determined after three washings with PBS (0.01 M, pH = 7.4). The excitation wavelength was 297 nm, and the emission wavelength was 527 nm.

#### 2.3.3. Zeta Potential and Membrane Depolarization Tests

The zeta potential was moderately modified as described by Bai et al. [32]. The logarithmic-phase bacterial fluid was centrifuged, and the precipitated bacterial cells were resuspended with a 40 mM HEPES buffer. Different concentrations of linalyl alcohol were added and incubated for 6 h. Finally, the zeta potential of each sample was measured using a laser particle size analyzer (Malvern Zetasizer Nano ZSE, Malvern Instruments, Malvern, UK).

The cells were precipitated and resuspended in PBS at an OD_600_ of 1.0 with a final concentration of 30 M of a 3.3-dioxide hydrosphere (DiOC_2_(3)) solution by spinning the logarithmic-phase bacterial fluid. Using a fluorescent enzyme labeling device (Infinite M200 PRO, TECAN, Nanjing, China), the cells were incubated for 5 min at room temperature before emitting fluorescence at a wavelength of 450 nm, with a fluorescence emission wavelength of 670 nm and an excitation wavelength of 450 nm [33].

#### 2.3.4. Determination of Chemical Configuration

FT-IR was employed to identify changes in the cell membrane composition [34]. The bacterial suspension was cultured for 6 h at 26 °C with linalyl alcohol (1 MIC or 2 MIC). The cells were collected and frozen before drying. Then, 2 mg and 100 mg of KBr were combined and ground. Without adding sterile water and 1% ethanol as a blank and control, spectra of each sample were generated using a Nicolet 10 FTIR spectrometer (Symmer Fisher Technology, Waltham, MA, USA). A total of 64 scans were carried out, and the spectra were recorded in the spectral range of 400 to 4000 cm^−1^ with a spectral resolution of 4 cm^−1^.

#### 2.3.5. Electron Microscope Analysis

Observations were carried out using a field-emission scanning electron microscope (FEGSEM) with a modified protocol described elsewhere [35]. *B. thermosphacta* cells were treated with linalyl alcohol, washed twice with PBS (pH 7.4), immersed in 2.5% glutaraldehyde overnight, washed twice with PBS, and dehydrated via graded dehydration with ethanol (30–100%). After lyophilization, the samples were fixed on the carrier stage with a conductive adhesive. A thin layer of gold was plated on the samples to observe the cell morphology using an FEGSEM (Thermo Scientific Verios G4 UC, Waltham, MA, USA).

To comprehensively analyze the changes in bacterial appearance and intracellular structure, negatively labeled cells were studied using a TEM [36]. Log-phase bacteria were centrifuged, washed, and resuspended in PBS (0.05 mol/L, pH 7.2) before use. Equal amounts of 2% phosphotungstic acid and cell suspension were used. The mixture was covered with a copper grid and dried for 15 min. The cell morphology was then observed under a TEM (Talos F200X G2, Thermo Scientific, Bremen, Germany).

### 2.4. Effect on Intracellular Substances

#### 2.4.1. Respiratory Chain Dehydration

In accordance with Sun et al. [37], 1 mL of the bacterial solution was taken in four sterile test tubes, and 2 mL each of a 0.05 mol/L Tris-HCl buffer (pH 8.6), 0.1 mol/L glucose solution, and 1 mg/mL 2,3,5-triphenyl tetrazolium chloride (red tetrazolium) solution were added and shaken thoroughly. Linalyl alcohol was added to the tubes and incubated in a shaker at a suitable temperature. Finally, two drops of concentrated sulfuric acid were added to terminate the reaction, and the color change was observed and photographed. Then, 5 mL of n-butanol was added to each tube, mixed well, and left for 30 min; afterwards, the upper layer of the liquid was centrifuged at 4500 r/min for 10 min, the absorbance value of the supernatant was measured at 490 nm, and the same amounts of sterile water and 1% ethanol were used as the blank control and negative control.

#### 2.4.2. *β*-Galactosidase and Alkaline Phosphatase

An amount of 10 mL of the bacterial solution was placed in a sterile test tube, 1 MIC and 2 MIC linalyl alcohol were added, and equal amounts of sterile water and 1% ethanol were used as blank and negative controls. The mixture was incubated at 26 °C for 6 h and then centrifuged at 6500 r/min for 15 min. Bacterial alkaline phosphatase was determined using the method of Zhao et al. [38], and the *β*-galactosidase content was determined using the method of cui et al. [39].

#### 2.4.3. SDS-PAGE of Proteins

In accordance with the instructions on the bacterial protein extract reagent box, 1 mL of the various treatments was pipetted, centrifuged, and washed several times with PBS, the precipitate was suspended in a cell lysis buffer, and 40 μL of PMSF and 80 μL of lysozymes were added, followed by incubation for 30 min at 37 °C. An amount of 20 mL of Dnase/Rnase was added, and incubation was continued for 10 min. Then, 20 μL DNase/RNase was added, followed by incubation for another 10 min. The extracts were examined via an SDS-PAGE electron magnetophoresis assay using a 10% concentration.

#### 2.4.4. Detection of the Intracellular Reactive Oxygen Species Level

The accumulation of reactive oxygen species (ROS) generated within cells was measured using the fluorescent dye 2,7-dichlorodihydrofluorescein diacetate (DCFH-DA). Bacterial fluid (5 mL) was taken into a sterile cardiovascular tube and treated with varying concentrations of linalyl alcohol for 2 h. The bacterial sediment was collected and suspended in a solution of DCFH-DA (20 μmol·L^−1^) (2 mL), followed by incubation in darkness for 30 min. The reaction mixture was then centrifuged, washed, and suspended in an appropriate volume of a PBS solution. The ROS fluorescence intensity of the different samples was measured using a fluorescence spectrophotometer (Infinite M200 PRO, TECAN, Nanjing, China) with excitation at 488 nm and emission at 525 nm [40].

#### 2.4.5. Linalyl Alcohol–DNA binding Assay

Bacterial genomic DNA was extracted from different treatments of bacterial genomic DNA using a Rapid Extraction Kit (Shanghai Jieri Bioengineering Co., Ltd., Shanghai, China). The concentration and purity of the DNA were assessed by measuring the absorbance at 260 nm and 280 nm, and the ratio of A_260 nm_/A_280 nm_ was more significant than 1.8. The DNA was electrophoresed for 45 min on 1.0% agarose gel at a voltage of 150 V and then stained with ethidium bromide (EB). 

#### 2.4.6. Spectral Measurement

An analysis of the relationship between linalyl alcohol and DNA was carried out using the fluorescence spectrum linking DNA to the two substances EB and DAPI [41,42]. Genomic DNA was diluted to 50 μg/mL in distilled water, and a 15 μg/mL EB solution was added for every 1 mL of the DNA solution. This was followed by incubation at 37 °C for 10 min in the dark. Different concentrations of linalyl alcohol were added, and incubation was continued at 37 °C for 30 min in a biochemical incubator. In addition, EB (1, 2, 4, and 8 μg/mL) was added to the linalyl alcohol–DNA complex and incubated at 37 °C for 30 min. An amount of 0.2 μg/mL of DAPI was added to the DNA and incubated for 30 min. Then, different concentrations of linalyl alcohol were added and incubated for 1 h at 37 °C. Finally, the fluorescence spectra were recorded separately (λex = 530 nm and λem = 560–700 nm; λex = 314 nm and λem = 380–650 nm).

### 2.5. Statistical Analyses

SPSS 24.0 software (SPSS Inc., Chicago, IL, USA) was used for the data analysis, and Origin 2023 software and BioRender (https://app.biorender.com/) were used for graphing. All experiments were carried out in triplicate, and the results are reported as the mean ± standard deviation. Duncan’s test was used to determine the significance (*p* < 0.05) for multiple comparisons. 

## 3. Results and Discussion

### 3.1. Minimum Inhibitory Concentration of Linalyl Alcohol

The minimum inhibitory concentration (MIC) is one of the leading indicators used to measure and evaluate the bacterial inhibitory ability of a substance. As shown in Table 1, the sterile water and 1% ethanol had almost no effect on bacterial growth. Linalyl alcohol showed significant antibacterial activity against the growth and multiplication of *B. thermosphacta*, and the antibacterial ability was positively correlated with the linalyl alcohol concentration. When the concentration of linalyl alcohol was 3 mL/L, no bacterial colonies were observed on the culture plates. Therefore, it can be concluded that the minimum inhibitory concentration (MIC) of linalyl alcohol for *B. thermosphacta* is 3 mL/L.

### 3.2. Effect of Linalyl Alcohol on the Growth Curve of B. thermosphacta

By measuring the OD value at 600 nm, the growth curve of *B. thermosphacta*, as depicted in Figure 1, was assessed to determine the effect of linalyl alcohol on the growth and reproduction of *B. thermosphacta*. *B. thermosphacta* in the blank control group and the ethanol group was in the delayed stage from 0 to 4 h, reached the logarithmic growth stage after 4 h, reached the maximum at 18 h, and reached the stable stage almost at the same time, indicating that 1% ethanol had no significant effect on bacterial growth. However, the bacterial liquid OD_600_ was much lower than that of the control group and remained almost constant when the concentration of linalyl alcohol was increased to 1 MIC. The experimental groups containing 1 MIC and 2 MIC linalyl alcohol showed significant differences in the growth curves, and the OD value and growth range were significantly lower than those of the blank group and negative control group, indicating slower bacterial growth. The results showed that linalyl alcohol delayed the growth cycle of *B. thermosphacta* cells and effectively inhibited the growth of *B. thermosphacta*. 

### 3.3. Mechanism of Linalyl Alcohol Cell Membrane Inhibition against B. Thermosphacta

#### 3.3.1. Surface Hydrophobicity Analysis

The surface hydrophobicity of bacterial cells is one of the important factors for bacterial growth and adhesion control. An increase in surface hydrophobicity means that bacteria can sink more easily, leading to a greater impact on bacterial growth [43]. From Figure 2, it can be observed that with an increase in the linalyl alcohol content and prolonged treatment time, the induced fluorescence intensity significantly increased. Additionally, there was a blue shift in the maximum fluorescence intensity, indicating an increase in the ANS fluorescence intensity and a change in the maximum fluorescence emission value. *B. thermosphacta* treated with linalyl alcohol exhibited a 1.34-fold and 1.59-fold increase in the ANS fluorescence intensity at the maximum fluorescence intensity compared to the untreated sample. The blue shift in the maximum emission value and the increase in the fluorescence intensity suggest that the linalyl alcohol treatment increased the hydrophobicity of *B. thermosphacta*’s surface, resulting in an increase in the number of hydrophobic regions binding with the ANS probe. This indicates that linalyl alcohol disrupts the cell wall of *B. thermosphacta*, exposing more hydrophobic phospholipid bilayer regions on the cell membrane. 

#### 3.3.2. Membrane Damage

##### Analysis of Crystal Violet and Fluorescein Diacetate Staining

The damage to the bacterial cell membrane was investigated via crystal violet staining, and fluorescein diacetate staining was used to verify whether the bacterial cells were damaged and then explore the mechanism of linalyl alcohol’s action on the cell membrane of *B. thermosphacta*. As shown in Figure 3A, the OD value of the experimental group treated with linalyl alcohol under crystal violet staining was significantly higher than that of the blank group and negative control group, indicating that linalyl alcohol promoted the uptake of crystal violet by the cells, resulting in an increase in the OD value at 590 nm compared with the untreated group. The results of the crystal violet staining experiments indicate that linalyl alcohol’s action could lead to cell membrane damage, thus increasing cell membrane permeability and enhancing its uptake of crystal violet.

Di-O-acetylfluorescein (FDA) can produce yellow-green fluorescein, and when the cell membrane is damaged, the intracellular fluorescence intensity decreases sharply [31]. As shown in Figure 3B, the fluorescence intensity of *B. thermosphacta* treated with linalyl alcohol was significantly lower than that of the untreated group, and the fluorescence intensity diminished with increasing concentration. The results indicate that different concentrations of linalyl alcohol can dramatically disrupt the bacterial cell membrane, leading to the loss of fluorescein in bacteria and thus reducing the fluorescence intensity of FDA. 

##### Zeta Potential and Membrane Depolarization

Reactions between the repressor and cell membrane lead to changes in the cell membrane potential. These interactions can be reflected by changes in the cellular zeta potential and transmembrane potential, thus revealing the mechanism of action of cell membrane damage [44]. Figure 4A shows the change in the surface zeta potential of heat-killed *B. thermosphacta*. The zeta potentials after the blank, control, 1 MIC, and 2 MIC treatments were −14.46, −13.67, −9.93, and −8.74 mv, respectively, indicating that the surface zeta potentials of the bacteria were significantly altered after treatment (*p* < 0.05).

Figure 4B shows the changes in the fluorescence intensity of *B. thermosphacta* after treatment with linalyl alcohol. The results show that treating *B. thermosphacta* with linalyl alcohol can lead to a substantial decrease in the membrane potential generated, indicating that linalyl alcohol can permeabilize the bacterial cell membranes and weaken their barrier function, which ultimately leads to the death of *B. thermosphacta*.

##### Analysis of Cell Membrane Composition

Figure 5 shows the changes in the cell membrane composition analyzed using Fourier-transform infrared spectroscopy (FT-IR). According to Lambert’s law, the transmittance is inversely proportional to the absorbent substance concentration, indicating that the smaller the transmittance value (the more significant the peak), the higher the substance concentration. From the figure, it can be seen that after treatment with linalyl alcohol, *B. thermosphacta* cells showed an increase in the bands around 2930 cm^−1^, indicating that linalyl alcohol increased the content of fatty acids in the cell membrane. Glycerophospholipids are the main lipids of membrane bilayers and play an important role in response to environmental stress. Tang et al. investigated the upregulation of bacterial glycerol-3-phosphate dehydrogenase (GPT1) after linalyl alcohol treatment, finding an increase in glycerol phosphate [45]. The bands around 1650 cm^−1^ and 1540 cm^−1^ correspond to proteins, and the intensity of the peaks increased, indicating that the protein content in the cell membrane of *B. thermosphacta* increased. This situation may be due to the damage to the cell membrane caused by linalyl alcohol, where the bacteria increase the synthesis of membrane proteins in response to such damage. The 1455 cm^−1^ wave number represents the bending of the -CH_2_ bond in lipids [46], which can affect the fluidity and stability of the membrane. The variation in the peaks around 1230 cm^−1^ and 1080 cm^−1^ can be attributed to the asymmetric and symmetric stretching vibrations of the phosphate molecule [47]; from the FT-IR graph, it can be seen that the peaks at both wave numbers increased at high concentrations, indicating that linalyl alcohol can affect the phosphate molecules in the bacterial cell membranes. The FT-IR results further confirmed that linalool induces cell membrane damage by affecting cell membrane fatty acids and membrane proteins, ultimately leading to cell death.

##### Morphological Changes in *B. thermosphacta*

After *B. thermosphacta* was treated with linalyl alcohol, the FEGSEM analysis showed that membrane damage had occurred. The membranes in the control and blank groups showed a normal morphology, including unbroken borders resembling smooth bars (Figure 6A,B). Unlike the control group, linalyl-alcohol-treated B. thermosphere showed bacterial rupture (Figure 6C,D). Transmission electron microscopy showed that the untreated cells were surrounded by rounded, unbroken areas (Figure 6E,F), showing the typical morphology of the cells. Unlike the non-treated group, the surface of the linalyl-alcohol-treated cells showed a collapse phenomenon, and there were traces of content outflow around the cells. These findings suggest that linalyl alcohol can impede growth by altering the morphological structure of the strain and increasing the intensity of damage in a dose-dependent manner.

#### 3.3.3. Study of the Effect of Intracellular Substances

##### Respiratory Chain Dehydrogenase

The respiratory chain is an energy source for maintaining regular cellular metabolism and transfers electrons through redox reactions from electron donors to electron acceptors. The tiny chemical TTC is reduced to red 1,3,5-tribenzoate (TF) when it interacts with respiratory chain dehydrogenase inside bacterial cells. Figure 7 depicts the impact of linalyl alcohol on *B. thermosphacta*’s respiratory chain dehydrogenase, which is clear from the color difference between the control and treatment groups.

It was demonstrated that treatment with linalyl alcohol significantly damaged the respiratory chain dehydrogenase in *B. thermosphacta* cells, and the color became more pronounced as the linalyl alcohol concentration was increased. Each group’s absorbance values in Figure 7 matched how each group’s colors changed. According to the findings, linalyl alcohol can stop *B. thermosphacta* from growing by upsetting the respiratory chain.

##### *β*-Galactosidase (*β*-GAL) and Alkaline Phosphatase (ALP)

Cell membrane permeability can be confirmed by the concentration of *β*-galactosidase and alkaline phosphatase [48]. *β*-GAL is a hydrolase located on the plasma membrane of bacterial cells and is the primary source of energy for bacteria; it catalyzes the hydrolysis of lactose to monosaccharides [49]. *β*-GAL decomposes p-nitrophenyl-*β*-D-galactopyranoside to produce p-nitrophenol. From the figure, it can be seen that the intracellular *β*-galactosidase exocytosis of *B. thermosphacta* treated with linalyl alcohol increased significantly in the bacterial suspension, with the contents rising by 31.08% and 40.09% compared with the control group. The results indicate that linalyl alcohol could disrupt the typical structure of the bacterium and cause the leakage of intracellular *β*-galactosidase, thus disrupting the normal physiological activities of the spoiled cells and hindering their growth.

ALP is a biological enzyme between the cell wall and the cell membrane of *B. thermosphacta*. When the cell wall is damaged, ALP will be detected with the outflow of material between the cell wall and the cell membrane; therefore, the permeability of the bacterium can be reflected by detecting the change in the ALP level. As can be seen from Figure 8, the ALP level was significantly higher in the treated group than in the untreated group, indicating that linalyl alcohol ruptured the bacterial cell walls, and the degree of rupture became more obvious with the increase in concentration, which is consistent with the study of Guo et al. [50].

##### SDS-PAGE of Proteins

The analysis of bacterial cell protein profiles is a relatively important indicator for bacterial strain studies. The SDS-PAGE analysis was used to determine whether there were differences in protein profiles between the treated and untreated samples. The protein bands of markers and cells are shown in Figure 9. The features range from 8 kDa to 180 kDa, and each bar represents the size of the protein. The untreated *B. thermosphacta* bands are significantly concentrated, while the treated cell bands have almost no large-molecule bands, with some accumulation of small-molecule bands. Such results suggest that linalyl alcohol may affect the synthesis of large-molecule proteins after linalyl alcohol treatment. Other studies found similar results for clove essential oil and litsea cubeba EO, which can affect protein synthesis and cause protein leakage [51,52].

##### Oxidative Damage to Cell Membranes

A non-fluorescent probe, DCFH-DA, was used to detect intracellular ROS levels. DCFH-DA itself is non-fluorescent and can freely cross the cell membrane; upon entering the cell, it can be hydrolyzed by intracellular esterases to generate DCFH, which cannot penetrate the cell membrane, thus allowing the probe to be readily reproduced in the cell. Oxidation of non-fluorescent DCFH by intracellular reactive oxygen species generates fluorescent DCF. As shown in Figure 10, the ROS levels in the *B. thermosphacta* cells increased with increasing linalyl alcohol concentration. The ROS levels in the *B. thermosphacta* cells in the blank and control samples were 201.66 ± 1.53 and 240 ± 1.73, respectively, while the bacterial fluorescence intensities of linalyl alcohol in the 1 MIC and 2 MIC treatments were 268.67 ± 1.53 and 520.0 ± 2.65, respectively. Therefore, linalyl alcohol treatment can attack and disrupt the cell membrane part of the lipid bilayer and induce intracellular ROS production, thus accelerating the death of *B. thermosphacta*. 

##### Effects of Linalyl Alcohol on Genomic DNA of *B. thermosphacta*

DNA damage can lead to bacterial death by interfering with gene expression and inhibiting the synthesis of enzymes and receptors. To determine the interaction between linalyl alcohol and the DNA of the target, DNA agarose gel electrophoresis, a competitive binding assay, was used. As shown in Figure 11A, the DNA bands were weakened after treatment with linalyl alcohol. The DNA bands (lanes 3 and 4) were discolored, suggesting that linalyl alcohol may promote DNA degradation or burst dye fluorescence. To ascertain whether linalyl alcohol destroys DNA and to further confirm the interaction between linalyl alcohol and DNA, a DNA competitive binding experiment between linalyl alcohol and EB and DAPI was carried out.

As shown in Figure 11, the fluorescence intensity gradually decreased with increasing doses of linalyl alcohol in the EB–DNA complex. This may be related to the insertion of EB into the DNA base pairs via competitive linalyl alcohol substitution, which reduces the DNA fluorescence intensity (Figure 11B). Meanwhile, in the DNA–linalyl alcohol complex system, the fluorescence intensity was significantly supplemented by the addition of EB. The fluorescence compensation effect was strengthened with the increase in the EB concentration (Figure 11C). The above results suggest that linalyl alcohol can bind to the DNA molecule in an EB-like manner but cannot degrade it.

Figure 11D shows the competitive inhibition of linalyl alcohol with the DAPI–DNA complex; the fluorescence of the DAPI–DNA complex after linalyl alcohol treatment was reduced, which indicates that linalyl alcohol can replace the binding site of DAPI and prefers to bind in the tiny grooves of DNA. Thus, in summary, the results suggest that linalyl alcohol can interact with DNA through intercalation binding and groove binding.

## 4. Conclusions

In conclusion, the current investigation demonstrates that linalyl alcohol has a substantial inhibitory effect against *Brochothrix thermosphacta*, mainly through the following mechanisms: First, linalyl alcohol alters the cellular structure, resulting in irreparable cell membrane damage. This finding might be confirmed using Fourier-transform infrared (FTIR) spectroscopy and the leakage of intracellular chemicals (*β*-GAL and ALP). Second, linalyl alcohol can prevent or speed up the breakdown of macromolecular proteins, depolarize the cell membrane, and increase intracellular ROS. Scanning electron microscopy and transmission electron microscopy observations further support this damage. In addition, linalyl alcohol can form hybrids with DNA, leading to changes in the DNA structure. The antibacterial action of linalyl alcohol may involve initially disrupting the physiological structure within the cells of the deteriorating bacteria, causing changes in cell contents and ultimately resulting in the rupture of the cell membrane and the death of the bacteria. The findings of this research contribute to the utilization of linalyl alcohol in addressing issues related to meat production and storage safety.

## Figures and Tables

**Figure 1 foods-13-00244-f001:**
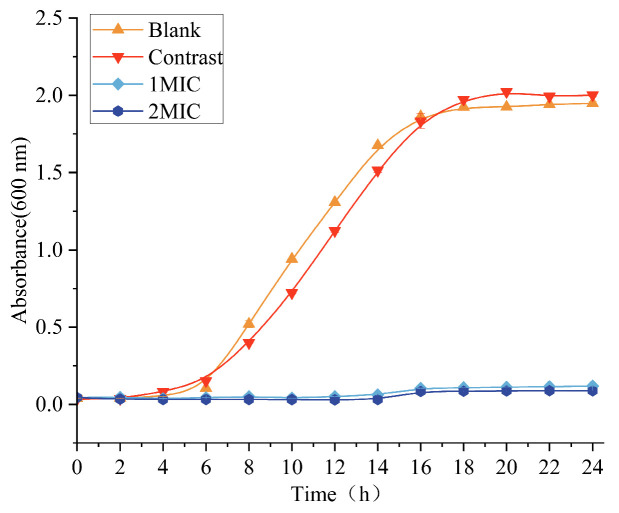
Effect of linalyl alcohol on the dynamic growth curve of *B. thermosphacta*.

**Figure 2 foods-13-00244-f002:**
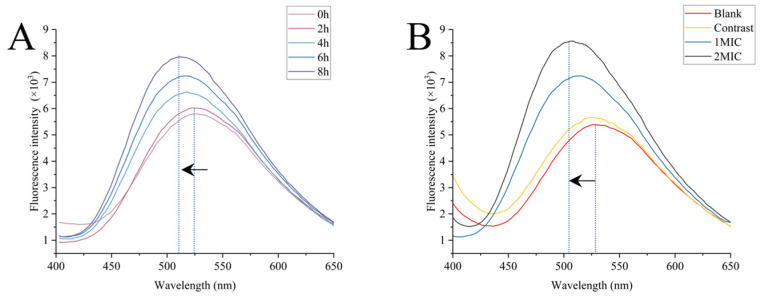
Effects of linalyl alcohol treatment on *B. thermosphacta* surface dehydration. Linalyl alcohol treated *B. thermosphacta* at different times (**A**); Treatment of *B. thermosphacta* with different concentrations of linalyl alcohol (**B**). The arrows in the figure indicate the direction the spectrum is moving.

**Figure 3 foods-13-00244-f003:**
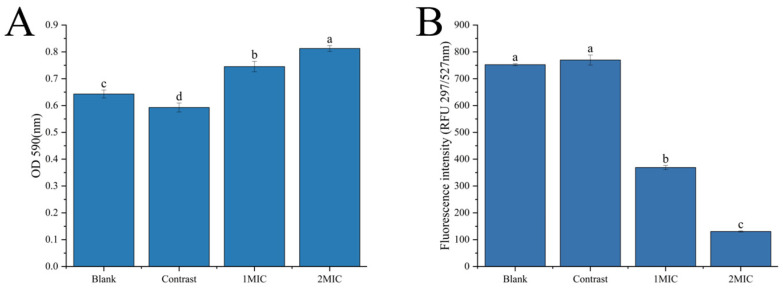
Changes in crystal violet absorption at OD_590_ (**A**), and changes in FDA fluorescence intensity (**B**). With in the same group, different lowercase superscript letters indicate significant differences (*p* < 0.05).

**Figure 4 foods-13-00244-f004:**
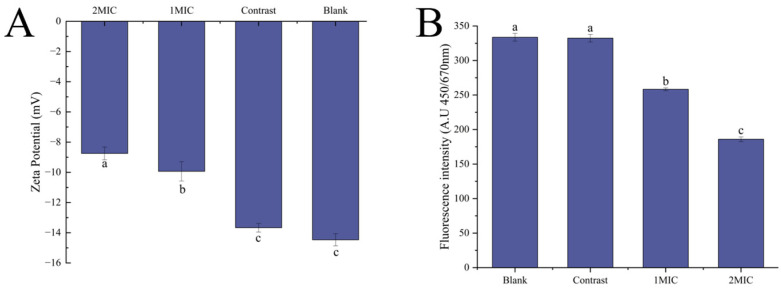
Linalyl alcohol’s depolarizing effects on *B. thermosphacta* zeta potential (**A**) and cell membrane fluorescence intensity (**B**). With in the same group, different lowercase superscript letters indicate significant differences (*p* < 0.05).

**Figure 5 foods-13-00244-f005:**
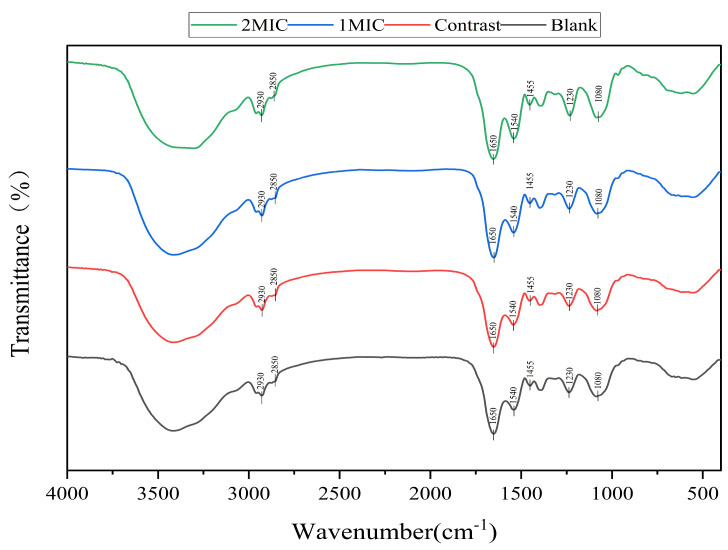
Linalyl alcohol analysis of the cell membrane composition of *B. thermosphacta*.

**Figure 6 foods-13-00244-f006:**
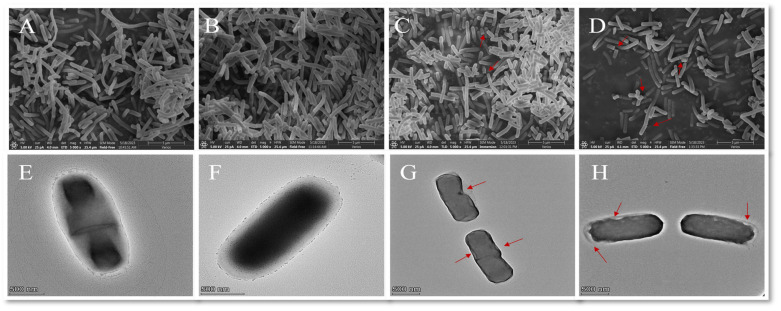
Field-emission scanning electron microscopy (**A**–**D**) and transmission electron microscopy (**E**–**H**) observations of the morphology of linalyl-alcohol-treated *B. thermosphacta* cells. (**A**,**E**): blank; (**B**,**F**): control; (**C**,**G**): 1 MIC linalyl alcohol; D and H: 2 MIC linalyl alcohol. The arrow indicates the damaged part.

**Figure 7 foods-13-00244-f007:**
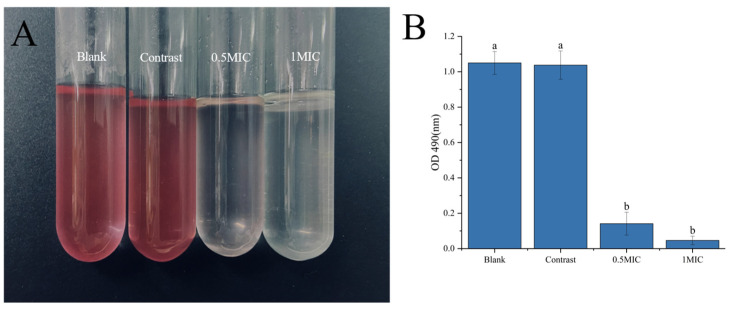
The effect of linalyl alcohol treatment (**A**), respiratory chain dehydrogenase (**B**). With in the same group, different lowercase superscript letters indicate significant differences (*p* < 0.05).

**Figure 8 foods-13-00244-f008:**
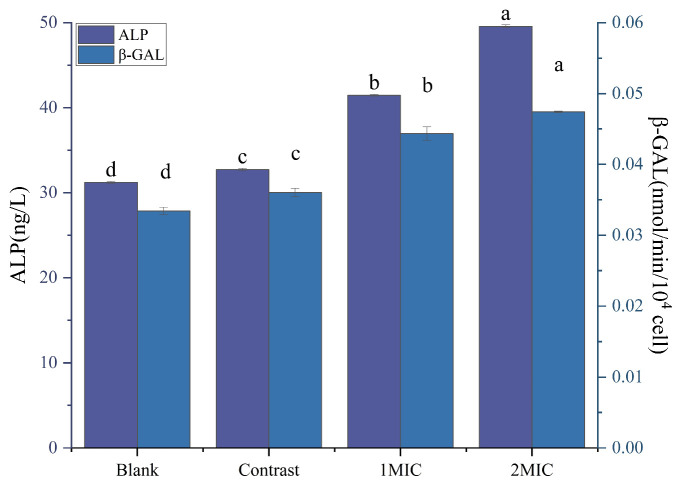
Effect of linalyl alcohol on *B. thermosphacta β*-GAL and ALP. With in the same group, different lowercase superscript letters indicate significant differences (*p* < 0.05).

**Figure 9 foods-13-00244-f009:**
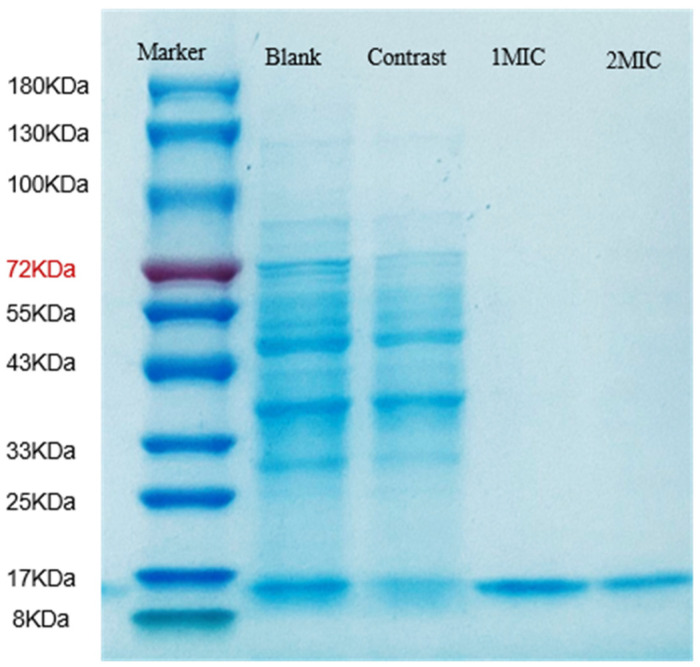
SDS-PAGE of *B. thermosphacta* protein treated with linalyl alcohol. Lane 1, blank; lane 2, control; lane 3, 1 MIC linalyl alcohol treatment; lane 4, 2 MIC linalyl alcohol treatment.

**Figure 10 foods-13-00244-f010:**
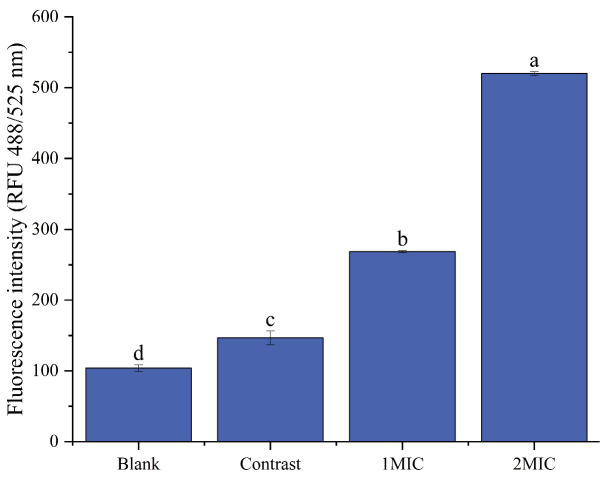
Effect of linalyl alcohol on *B. thermosphacta* ROS fluorescence intensity. With in the same group, different lowercase superscript letters indicate significant differences (*p* < 0.05).

**Figure 11 foods-13-00244-f011:**
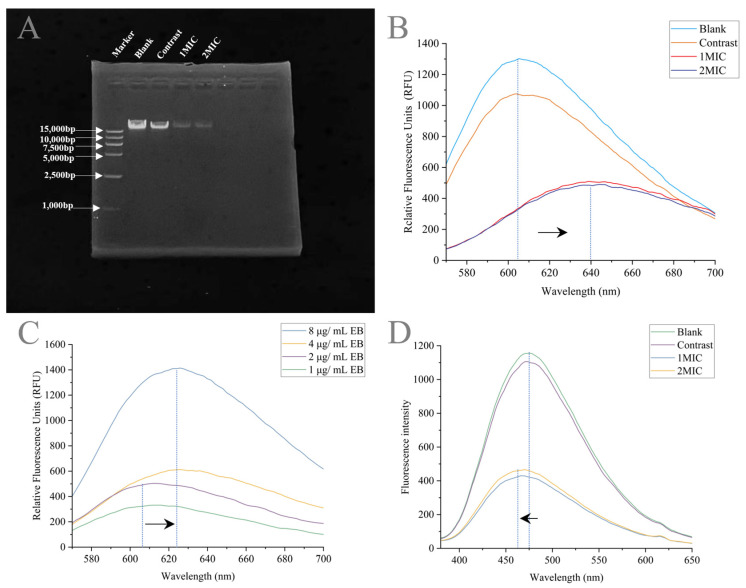
(**A**) Gel retention experiments. (**B**) Fluorescence spectra of EB–DNA complexes treated with different concentrations of linalyl alcohol. (**C**) Fluorescence spectra of linalyl-alcohol–DNA complexes treated with varying concentrations of EB. (**D**) Fluorescence spectra of DAPI–DNA complexes treated with different concentrations of linalyl alcohol. The arrows in the figure indicate the direction the spectrum is moving.

**Table 1 foods-13-00244-t001:** Minimum inhibitory concentration (MIC) of linalyl alcohol against *B.thermosphacta*.

Bacteria for Test	Control	1% Ethanol	Linalyl Alcohol Concentration (mL/L)					
			0.375	0.75	1.5	3	6	12
*B. thermosphacta*	+++	+++	+++	+++	+++	−	−	−

Notes: “+++” represents a large number of colonies, and “−” indicates no colonies.

## Data Availability

The data used to support the findings of this study can be made available by the corresponding author upon request.

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
