# Peer review of "Studies on the Inhibition Mechanism of Linalyl Alcohol against the Spoilage Microorganism *Brochothrix thermosphacta"

_foods, 2024, doi:10.3390/foods13020244_

Round 1
Reviewer 1 Report
Comments and Suggestions for Authors
The manuscript entitled Studies on the inhibition mechanism of linalyl alcohol against the spoilage microorganism Brochothrix thermosphacta was thoroughly check. here are some suggestions for authors which they need to address.
In abstract
1. clearly mention the determined minimum inhibitory concentration (MIC) of linalyl alcohol against B. thermosphacta.
2. Divide the paragraph into smaller sentences to improve readability and clarity.
3. Highlight the mechanism by which linalyl alcohol increases the fatty acid content of the cell membrane to resist damage.
Introduction
1. The first two sentences of the introduction part rise many concerns, I don't think so, revised the starting sentences of the introduction. start with logical sentences.
2. Authors mentioned that linalyl alcohol has been successfully employed in other studies, but did not mention then why they choose this study, any further innovation?
3. The introduction part is lack of aims and objectives of this study, after adding this the point number 2 would be tickle.
4. Add few more details about health concern of such study.
Methods and materials
1. why authors did surface hydrophobicity? just for data quantity? justify in methods and material section.
2. minimum inhibition effect should explain in detail for better reproducing by other authors
3. 2.4.2 this section need revision, hard to understand what the actual protocols authors was followed.
4. same for 2.4.3, the English language is not good, hard to understand, demonstrate the experiment in simple way
Results and discussion
1. First all of the figures have poor quality, provide clear and concise figures.
2. Highlight the important changes in figure 2 with adding arrows.
3. Authors used significant difference software, however no signific letters nor anything mentioned in figure 3.
4. same for figure 4.
5. Clarify that a lower transmittance value indicates a higher concentration of substances.
6. Describe the changes observed in the cell membrane composition after treatment with linalyl alcohol.
7. Highlight the bands at 1650 cm-1 and 1540 cm-1, which represent proteins, and explain that their intensity increased, indicating an increase in protein content in the cell membrane.
8. Mention the significance of the peak at 1455 cm-1, which represents the bending of the -CH2 bond in lipids
9. Change the color of bar chat of the figure 8
10. Lable each of the lane in SDS-PAGE figure, better to provide bit brighter photo of gel
11. Indicate the important shifts in figure 11 with adding arrows.
Conclusion
Revise the conclusion and mold it toward the basic findings of this research, add some application of this study.
Author Response
Dear reviewer, we have replied to the review comments in the attachment.

Reviewer 2 Report
Comments and Suggestions for Authors
The idea behind this article is very interesting: a saprophytic strain of bacteria that increasingly spoils packaged products and raw materials that have been treated with one of the pure ingredients of the oils. A very timely topic and consistent with the use of natural antimicrobial substances, and the effects have been studied using many new methods. To make the manuscript more valuable, it needs to be changed a bit and some inconsistencies/unknowns should be clarified.
I have included my comments on the text below:
1. The authors write the names of microorganisms in regular font throughout the text. This is an error because the names of microorganisms are written in italics. Please correct this throughout the text.
2. I also have a comment about the use of abbreviations as a whole. The rule is that when using an abbreviation for the first time in a text, it should be explained after its first use in the text. It's very different here: line 73 - "NA medium"; line 85 - "LB liquid medium"; line 133 - "Observations were made using FEGSEM"; line 114 - "HEPES"; line 266 - "FDA" and many others. Sometimes it is practiced to list all the abbreviations used in the article - this makes reading easier.
3. Line 47 and 48 - it is written "anti-Schistosoma mansioni" rather than "anti-Schistosoma mansion".
4. The "Materials and methods" chapter is written as a description of actions performed (past tense). Here the Authors use language that is more of a "take and add" recipe rather than a "taken and added" recipe. At this point, a language correction will be needed.
5. The "Materials and methods" chapter should contain the names of device manufacturers, but in many places they were missing.
6. When specifying the origin of a bacterial strain, provide their name (line 71) and do not place it in the subsection called "Reagents" - this is living matter!
7. I have doubts about the MIC determination methodology used. This is usually determined using the liquid medium method and assessing the presence of turbidity. Based on the results obtained in the MIC method, the number of bacteria present in the above fluid is determined using the plate method (from the MIC determination). And then we determine the minimum killing concentration of the preparation/compound. I also do not understand why the authors use the values "1 MIC" or "2 MIC" in the further text.
8. I have a question about the wavelength for OD measurement - sometimes it is OD 600 nm (e.g. line 91), other times it is 590 nm (line 101). What are the causes of these differences in the wavelength used?
9. Subchapter 3.2. - the sentence "This indicates that l inalyl alcohol can slow B. thermosphacta's development and even cause it to pass away over a shorter period...." - I do not agree with this sentence, because in order to claim this, the minimum concentration should be determined lethal test alcohol (MBC).
10. I am puzzled by the inconsistency in the drawings and text. So in the text I found the following information: "Figure 4A shows the change in the surface zeta potential of heat killed S. thermophilus" (lines 280-281), while the caption under this figure (lines 280-281) reads as follows: "Figure 4 "Linalyl alcohol 's de polarizing effects on B. thermosphacta Zeta electrodes and membranes". Where does the S. thermophilus strain come from? Similarly, the authors write about this strain in line 350.
11. Figure 7 - is signed incorrectly because there are no explanations under the figure as shown in Figures 7A and 7B. Where do the results for 0.5 MIC come from (?!), if in the rest of the text they are presented for 1 MIC and 2 MIC?
12. Figure 9 - here the caption informs about "Effect of linalyl alcohol on the proteins of Mycobacterium rot" (line 377). Isn't this a mistake?
13. The discussion is not very extensive and the results obtained in similar areas by other authors should be more strongly emphasized.
14. It is worth reading other literature related to bacteria of the Brochothrix genus - for example by Prof. Agnieszka Nowak from the Lodz University of Technology (Poland).
Author Response
尊敬的审稿人,我们已回复附件中的审稿意见。

Reviewer 3 Report
Comments and Suggestions for Authors
The authors analyze the Studies on the inhibition mechanism of linalyl alcohol against two the spoilage microorganism Brochothrix thermosphacta. The experiment and the discussion are pretty well prepared. Unfortunately, the paper has to be majorly revised.
Here are some remarks:
1. The paper has some typos and minor editing errors. There is much missing space in the text and some double spaces. I highlighted them in yellow in the manuscript file.
2. The name of the bacteria should be written in italics and capital letters.
3. The authors should highlight the novelty of their work.
4. Please add some relevant references to the topic of the paper.
5. There must be a citation in the last paragraph of the Introduction.
6. The numbering of the Introduction, and Materials and method is missing.
7. The material and method section should be written in past perfect tense.
8. Table 1: The table is misleading. Sterile water and 1% ethanol solution were used as a reference for linalyl alcohol; please mark this in the table; for example, put these values in an additional row.
9. Table 1: what was the number of colonies at 2 and 2.5 mL/L concentration?
10. FT-IR can be used for quantitative testing if the calibration curve is prepared. Has something like this been done? Please attach the results.
11. Please provide the FT-IR spectrum in the same style as other figures.

English is acceptable, but there are a lot of typos and editorial mistakes which make the paper hard to read.
Author Response
尊敬的审稿人,我们已回复附件中的审稿意见。

Reviewer 4 Report
Comments and Suggestions for Authors
The manuscript entitled “Studies on the inhibition mechanism of linalyl alcohol against the spoilage microorganism Brochothrix thermosphacta” is well written and well structured by the Authors in each of its sections and denotes considerable scientific and methodological rigor.
The graphical abstract is very explanatory and clarifies at a glance the theme explored in the manuscript.
The introduction is well balanced, relevant and adequately accompanied by an updated bibliography. However, I remind the Authors to always write the genus and species of the bacterium in italics.
The same thing should be considered throughout the text because it is incorrect in several places. Furthermore, paragraph 2.3.3. it should start with a capital letter.
In the note to table 1 the Authors should indicate a range when referring to a high number of colonies, more colonies and a medium number of colonies also to give a more precise reference to the reader.
The discussion is very clear and in-depth and does not require further intervention in my opinion.
Author Response
尊敬的审稿人,我们已回复附件中的审稿意见。

Round 2
Reviewer 1 Report
Comments and Suggestions for Authors
Authors endorsed all of the suggestions. No further comment from here
Reviewer 3 Report
Comments and Suggestions for Authors
The authors have improved their paper according to the reviewer's recommendations. Before publishing, please correct some typos.
Comments on the Quality of English LanguageThe English is ok